# Digital Classification of Chilean Pelagic Species in Fishing Landing Lines

**DOI:** 10.3390/s23198163

**Published:** 2023-09-29

**Authors:** Vincenzo Caro Fuentes, Ariel Torres, Danny Luarte, Jorge E. Pezoa, Sebastián E. Godoy, Sergio N. Torres, Mauricio A. Urbina

**Affiliations:** 1Departamento de Ingeniería Eléctrica, Facultad de Ingeniería, Universidad de Concepción, Concepción 4070371, Chile; vicaro@udec.cl (V.C.F.); aritorres@udec.cl (A.T.); dannyluarte@udec.cl (D.L.); jpezoa@die.udec.cl (J.E.P.); segodoy@udec.cl (S.E.G.); sertorre@udec.cl (S.N.T.); 2Departamento de Zoología, Facultad de Ciencias Naturales y Oceanográficas, Universidad de Concepción, Concepción 4070386, Chile; 3Instituto Milenio de Oceanografía (IMO), Universidad de Concepción, Concepción 4070386, Chile

**Keywords:** fish recognition, remote sensing, industrial sensors, object detection, computer vision

## Abstract

Fishing landings in Chile are inspected to control fisheries that are subject to catch quotas. The control process is not easy since the volumes extracted are large and the numbers of landings and artisan shipowners are high. Moreover, the number of inspectors is limited, and a non-automated method is utilized that normally requires months of training. In this work, we propose, design, and implement an automated fish landing control system. The system consists of a custom gate with a camera array and controlled illumination that performs automatic video acquisition once the fish landing starts. The imagery is sent to the cloud in real time and processed by a custom-designed detection algorithm based on deep convolutional networks. The detection algorithm identifies and classifies different pelagic species in real time, and it has been tuned to identify the specific species found in landings of two fishing industries in the Biobío region in Chile. A web-based industrial software was also developed to display a list of fish detections, record relevant statistical summaries, and create landing reports in a user interface. All the records are stored in the cloud for future analyses and possible Chilean government audits. The system can automatically, remotely, and continuously identify and classify the following species: anchovy, jack mackerel, jumbo squid, mackerel, sardine, and snoek, considerably outperforming the current manual procedure.

## 1. Introduction

In 2020, the Food and Agriculture Organization (FAO) estimated the global production of aquatic animals to be 178 million tons, recognizing the essential contribution of the fisheries and aquaculture sectors to global food security and nutrition [1]. Thus, the management and sustainability of fishing stocks are crucial to ensure the protection of hydrobiological resources and their own environment, where illegal, unreported, and unregulated (IUU) fishing negatively impacts many developing and under-developed countries [2]. Chile is not exempt from these problems, aggravating the situation mainly due to overfishing.

To prevent a critical condition, the monitoring and control of fish landings and catch quotas in Chile is overseen by the Fisheries and Aquaculture National Service (Sernapesca). Sernapesca’s role is to execute the national fisheries policy and oversee its compliance, and, especially, to ensure the proper application of legal and regulatory norms regarding fishing, maritime hunting, and other forms of exploitation of hydrobiological resources [3]. Contrary to expectations, Sernapesca faces significant challenges due to its limited resources and the vast number of artisan vessels, fish species, and fish volumes in the Humboldt current system. The artisan fleet comprises 12,384 vessels and lands more than 900,000 tons of fish and jumbo squid per year [4]. To date, the inspection process relies on manual methods, which restricts its efficiency to the expertise of certifying officers and uses a small sample size (between 20 and 60 kg of fish per landing). Therefore, the implementation of new technology could play a significant role in improving Sernapesca’s inspection system. This technology could also help other national industries such as the forestry industry by measuring the dimensions and characteristics of logs and the mining industry by identifying mineral compositions in a conveyor belt, among others.

Focusing on the main challenges and research gaps that need to be addressed in the field of smart aquaculture, such as recognition, measurement, tracking, and classification, a couple of studies have presented overviews of proposed techniques and approaches by bridging the gap between solutions developed in academia and the practical requirements of the fishing industry [5,6]. For instance, deep learning techniques have emerged as a promising solution that enables feature extraction and designing of models capable of automatically identifying and classifying fish species in digital images within a broad range of environments [7,8,9,10]. Yu et al. proposed a scheme that utilizes Mask R-CNN [11] to segment fish images and measure their morphological features, eliminating the need for manual operations and reducing subjectivity while designing a robust model for both pure and complex backgrounds [12]. Furthermore, deep learning models have been applied in environmental monitoring, demonstrating their effectiveness in extracting ecologically useful information from video footage, such as detecting fish species and quantifying fish abundance in different habitats [13,14]. To improve the performance of deep learning models in fish recognition, various techniques have been explored. One study focused on fine-tuning and optimizing the VGG16 network [15] for classifying specific fish species found in Verde Island, achieving significant accuracy improvements through data augmentation and model refinement [16]. Another study addressed the limitations of existing fish recognition systems, emphasizing the need to include issues such as arbitrary fish sizes and orientations, feature variability, environmental changes, poor image quality, and segmentation failures for accurate classification [17].

Additionally, there are some further investigations that align closely with the objectives of this paper. Patent number CN214282791U presented a system that identifies and counts fish species passing through a water tube using a camera [18]. Patent number WO2020023467A1 presented a system that identifies fish using underwater cameras, similar to patent number CN214282791U, which uses convolutional neural networks [19]. Patent number CN111738139A resembles our solution in the sense that it also performs fish detection using the YOLO model trained on dead fish and a graphical interface to monitor its operation. Nonetheless, it differs from ours in its application and in the fish species on which it was trained (fish species found in cultivation ponds) [20]. Finally, patent number CN112949408A presented one of the most similar systems to our proposed solution. It combines real-time detection based on the YOLO model and classification with convolutional neural networks. However, it differs in the detected species, which are alive in a corridor [21].

In this paper, we present a fish recognition system based on deep convolutional neural networks. The proposed system can automatically, remotely, and continuously identify and classify different Chilean pelagic species in real time, as well as estimate their size. It considerably outperforms the current manual procedure carried out by the Sernapesca inspectors.

The rest of this paper is organized as follows. In Section 2, a detailed description of the proposed system is provided. In Section 3, the main results are presented and discussed. Finally, in Section 4, we present our conclusions and future work.

## 2. Materials and Methods

The proposed system for the detection, classification, and size estimation of pelagic species is composed of seven main modules, including both hardware and software components. The architecture of the system is illustrated in Figure 1. The following subsections provide detailed descriptions of each system module. Section 2.1 describes the hardware setup for the “Images Captured by IP Cameras”module. Section 2.2 describes the data collection process corresponding to the “Image Labeling” and “Training Preprocessing” modules. Section 2.3 describes the preprocessing algorithms performed prior to detection, corresponding to the “Inference Preprocessing” module. Section 2.4 and Section 2.5 describe the training and evaluation procedures of the detection and classification models, corresponding to the “Fish Discrimination” block. Finally, Section 2.6 and Section 2.7 describe the weight estimation and the web user interface, corresponding to the “Length and Weight Estimation” and the “Web Interface” modules, respectively. Section 2.7 also includes the description of the “Postprocessing” module that is executed after the detection and before the data are uploaded to the web user interface.

### 2.1. Hardware

To implement the fish recognition system installed on a conveyor belt in fishing plants, an optomechanical device was designed. The main component of the device is a gate that consists of a stainless steel structure, two RGB cameras, flicker-free LED panels to avoid visual flickering problems in the cameras, and a ventilation system.

Two versions of the gate were designed and implemented. The first version has four LED panels distributed around two Hikvision 2MP IP cameras (model DS-2CD2721G0-IZS), which operate in the visible spectrum. The connection to the cameras can be established through HTTP and RTSP protocols, and management and monitoring can be performed using Hikvision’s control software. This software allows the extraction of video feeds with a variable resolution, capture rate, bitrate, and other user-modifiable characteristics. The model selected for the cameras in the design of the second gate was the Hikvision DS-2CD1653G0-IZ. This 5MP model can operate in low-light conditions, and the field of view can be conveniently zoomed in or out according to the size of the species to be classified. This version of the gate has a single LED panel located in between the two cameras. The first design version of the gate was installed in an industrial fishing plant in south-central Chile, which we named Plant 1, whereas the second design version of the gate was installed in a different plant with conditions similar to those of Plant 1, which we named Plant 2, also in south-central Chile. Figure 2 shows a representation of the gate installed in Plant 1, showing the stainless steel structure mounted on a support beam, along with the lighting system with 4 LED panels, and the ventilation system. In contrast, Figure 3 displays a schematic view of the gate installed in Plant 2, highlighting the presence of a single LED panel positioned at the bottom of the structure.

### 2.2. Data Collection

Data capture was performed remotely by establishing a direct connection from an external computer to the cameras installed on each gate. For Plant 1, the captures consisted of images with a resolution of 1920×1080 pixels taken at a fixed frame rate of 8 frames per second (FPS). This frame rate ensured synchronization between the conveyor belt’s speed and the fish’s movement, avoiding any potential duplications. Data were captured from April 2021 to July 2021, encompassing morning, afternoon, and evening sessions. Similarly, for Plant 2, the images captured had a resolution of 1920×1080 pixels, but at a fixed frame rate of 1 FPS due to the plant’s slower conveyor belt speed. Captures were made from November 2022 to March 2023, also contemplating varied days and schedules. Considering the substantial data volume, data capture was not performed continuously for extended periods of time to minimize storage requirements.

The captures were manually reviewed for the presence of accompanying fauna, which were caught together with the target species and can either be exploited or discarded. The free access platform, Roboflow, was used to tag the images [22]. Expert Sernapesca certifiers also participated in this process. The labeling process involved creating polygon-shaped masks to identify and segment every pelagic species instance within an image. The samples considered were divided into plants, with the species anchovy (*Engraulis ringens*), jack mackerel (*Trachurus murphyi*), mackerel (*Scomber japonicus*), and sardine (*Strangomera bentincki*) belonging to Plant 1, and jack mackerel, mackerel, jumbo squid (*Dosidicus gigas*), and snoek (*Thyrsites atun*) belonging to Plant 2. The final number of labeled samples for each species and for each plant is summarized in Table 1. Future work will consider the installation of more gates in additional industrial plants and the inclusion of more marine species in the database.

To address the challenge of identifying and labeling fish species that were overlapping or present in large numbers within the same image, two image processing techniques were used. The first technique involved applying blurring to the image background while preserving the labeled species, effectively removing unidentified spots. The second technique randomly selected an image from a collection of several pictures showing sections of an empty belt, replacing areas of the original image that were not labeled. In this way, the unlabeled fish were removed, and by replacing parts of the image with a segment of the empty belt, it allowed the model to learn the differences between a fish and an empty belt. These techniques were randomly applied with an equal probability for each image in the database. Examples of modified images using these techniques are shown in Figure 4, illustrating the blurring effect in Figure 4a, and the replacement of unlabeled areas with an image of the empty belt in Figure 4b.

Finally, data augmentation techniques were applied to improve the model’s ability to detect fish, and, especially, to recognize the accompanying fauna, which has a low probability of occurrence in a common fish landing. The data augmentation techniques used included horizontal and vertical flips, maximum 30-degree rotations, brightness and contrast changes, noise addition, and blurring.

### 2.3. Inference Preprocessing Module

The inference preprocessing module consists of three main functions that are applied prior to the detection and classification models. Firstly, the color composition format of the images is transformed from BGR to RGB. This conversion is necessary because OpenCV, the image processing library, uses BGR format. Secondly, the images are resized using a letterboxing algorithm, ensuring that the original aspect ratio is preserved while adjusting the resolution from 1920×1080 to 640×640. Lastly, the preprocessing module includes a function to normalize the pixel intensity values of the images from the original range of 0 to 255 to the range of 0 to 1.

### 2.4. Model Training

For the development of the fish detection and classification system, several convolutional neural network models were utilized, including YOLOv4, YOLOv7, and Mask R-CNN. The choice of these models was based on the distinctive features of their architectures and their specific capabilities to detect objects in images.

YOLOv4, which stands for ’You Only Look Once’, is an object detection model that uses bounding boxes to tightly enclose and associate a class label for objects of interest within an image, all within a single-step neural network architecture (i.e., the input image is processed in one pass). This model was initially chosen for its high inference speed, which is particularly well suited for real-time object detection and processing. However, due to its reliance on bounding boxes, YOLOv4 faced challenges in accurately classifying fish species with overlapping features.

Mask R-CNN focuses on a different approach known as instance segmentation. This model goes beyond bounding boxes and provides detailed object segmentation, creating masks or polygons, which are pixel-level structures that delineate the exact shape of each detected object. This segmentation approach offers rich and detailed information about the objects, which can be especially valuable when dealing with fish species with overlapping appearances. However, its inference speed is considerably slower than any YOLO network, reaching rates of 1 to 2 FPS using an Nvidia A100 GPU in Google Colab [23].

With the release of YOLOv7, a new model emerged that combined the benefits of obtaining mask-like information similar to Mask R-CNN but with a significantly faster inference speed that outperformed YOLOv4. Consequently, YOLOv7 emerged as the most suitable choice among the evaluated models [24].

The hyperparameters employed for the YOLOv7 model were as follows: an initial learning rate of 0.01, an image size of 640×640 pixels, a maximum epoch limit of 500, an intersection over union (IOU) threshold of 0.2, a batch size of 16, and the stochastic gradient descent (SGD) method utilized as the optimizer. In addition to the aforementioned hyperparameters, early stopping was implemented during the model training process. This technique allowed the training to terminate when the model ceased to make further progress after a predetermined number of epochs, specifically set at 100. The database was divided, allocating 70% of the images for training purposes and reserving the remaining 30% for validation.

### 2.5. Model Evaluation

The performance of the model was measured using the following metrics: precision, confusion matrix, macro-average precision, and mean average precision. The latter two are explained in this subsection.

#### 2.5.1. Macro-Average Precision (MP)

The macro-average precision is the average precision calculated across all classes or categories in a multi-class classification problem. It gives equal weight to each class without considering class imbalance. The macro-average precision is calculated by taking the mean of the precision values for each class.
(1)$$MP=(P_{C1}+P_{C2}+\cdots +P_{CN})/N$$
where $P_{CN}$ is the precision for class N, and N is the total number of classes. It is also the mean of all individual class precision values reported in the main diagonal of a normalized confusion matrix.

#### 2.5.2. Mean Average Precision (mAP)

The mean average precision is a commonly used performance metric in object detection tasks. It measures the average precision across multiple classes and different levels of prediction confidence thresholds. The mAP summarizes the precision–recall curve by calculating the area under the curve (AUC) and provides an overall performance score. It is calculated as the mean of the average precision (AP) values for each class.
(2)$$mAP=(AP_{C1}+AP_{C2}+\cdots +AP_{CN})/N$$
where ${AP}_{CN}$ is the average precision for class N, and N is the total number of classes.

### 2.6. Length and Weight Estimation Module

The weight of each species was estimated through a relationship with its length given by
(3)$$W=aL^{b}\;.$$

Equation (Equation 3) presents the relationship between the fish weight, *W*, in grams, and the fish standard length, *L*, in centimeters, incorporating two parameters denoted as *a* and *b*. It is important to note that these parameters can vary depending on factors such as the geographical location, season, and other variables specific to each fish species. Consequently, the estimations may differ from year to year and across fishing grounds. Table 2 displays the adjusted parameters for four fish species. The data points used to fit Equation (Equation 3) were determined through laboratory measurements. To implement size estimation on the captured images, the width and height of the pixels within the respective bounding boxes were utilized. Moreover, for the size estimation, the total length of the fish, approximately represented by the diagonal of the bounding box, was adopted instead of the standard length. This modification was necessary due to the inherent difficulty in accurately identifying the hypural crease (the tail end) within a bounding box, which serves as the basis for calculating the standard length. By considering the total length, which accounts for the overall size of the fish captured in the image, a more accurate estimation of the weight can be obtained in a wider range of scenarios. Additionally, a conversion factor was applied to obtain the length value in centimeters, which corresponds to the actual distance represented by a single pixel in the real world. This conversion factor was determined in advance.

### 2.7. Software Interface

To implement the processing algorithms related to the proposed system, a web user interface was developed based on the Laravel Framework, which incorporated a number of benefits in terms of security that are relevant to industrial environments. Additionally, Python 3.7.x was utilized for programming the preprocessing, postprocessing, and sorting algorithms.

In this interface, an algorithm running on a server continuously monitors the cameras installed on the gates. It utilizes a CPU version of the YOLOv7 model specifically trained for the corresponding plant to automatically detect the presence of fish. The inference information, which includes a timestamp, the length, weight, model used, gate name, plant name, and other pertinent details, is then uploaded to a separate server’s database. Figure 5 illustrates this process, showing the flow of information from the detection algorithm to the database.

The postprocessing algorithm incorporates three modules. The first module is responsible for discarding detections based on their size, allowing for more precise filtering of the results. The second module implements a multiclass non-maximum suppression algorithm (NMS), which helps eliminate redundant or overlapping detections. Lastly, the third module is responsible for drawing the resulting bounding boxes of each detection on the original image. It also includes the predicted label and confidence level, providing visual representations and informative annotations. Explained in more detail, the module for discarding detections specifically focuses on two scenarios: when a sardine or an anchovy exceeds the threshold of the average size (i.e., larger than 20 cm) and when a jack mackerel or mackerel falls below the threshold of the average size (i.e., smaller than 20 cm). By identifying these instances, the module effectively filters out detections that meet these size conditions for both cases, ensuring that only fish within the expected size ranges are considered valid. By discarding abnormal detections, the system improves the accuracy and reliability of the subsequent processing and analysis steps.

The module for the multiclass NMS plays a crucial role in optimizing the detection results. It iterates over all the bounding boxes detected and checks for close proximity between boxes of different classes. If two bounding boxes are found to be very close to each other, the module eliminates the box with the lower confidence level. This step ensures that only the most confident and distinct detections are retained, preventing redundancy and improving the overall accuracy of the system.

Once the valid bounding boxes have been determined by the previous modules, they are visualized in the original image using predefined unique colors assigned to each fish class. Additionally, the classification model’s confidence level for each detection is displayed alongside the bounding box and label, providing a clear and intuitive understanding of each detected fish species.

Finally, the system includes a data storage module that facilitates the organization and management of the processed data. This module includes the delivery of the estimated size and weight of each detected fish, along with a comprehensive summary of all processed data. The system generates files in Excel and PDF formats, allowing for convenient access and analysis of the collected information. In the web user interface, users have the option to select specific fish landings and view detailed reports containing statistics and graphs related to the processed data.

## 3. Results and Discussion

The trained models, as described in Section 2.4, were evaluated on the validation dataset, corresponding to 30% of the labeled data that were not used for the training adjustments, which were obtained from the database of the two industrial plants, as summarized in Table 1. In the case of Plant 1, since it was the first plant where the system was implemented, there was an additional manual validation conducted by Sernapesca inspectors. They analyzed the landing videos and assisted in tasks such as species identification and correction of erroneous detections from preliminary versions of the models, which later allowed adjusting the final model for this plant.

### 3.1. Results for Plant 1

For Plant 1, the model was evaluated on 752 fish samples. The results obtained are summarized in the confusion matrix in Figure 6. The system achieved accuracy values of 0.9 for anchovy, 0.93 for jack mackerel, 0.95 for mackerel, and 0.83 for sardine, resulting in a value of 0.9 for the MP. The highest false positive (FP) background precision value, which corresponds to background elements erroneously detected as belonging to one of the trained classes, was 0.48 for jack mackerel. Furthermore, the highest false negative (FN) background precision value, which corresponds to species that were not detected by the model and were mistakenly classified as background, was 0.16 for sardine. There were some instances of misclassification, with jack mackerel misclassified as mackerel at a rate of 0.04 and anchovy misclassified as sardine, and vice versa, at a rate of 0.01.

Figure 7a showcases the progress of both the loss function and mAP throughout the training phase for Plant 1. The red dot represents epoch 137, which corresponds to the weights used with the model to achieve the best results, where the maximum mAP value of 0.838 was obtained. These weights were utilized in subsequent stages during the inference process with the model. Figure 7b shows the precision–recall curve for the masks generated by the YOLOv7 algorithm during the validation process in Plant 1. This curve showcases the performance of the model in terms of precision and recall for different threshold values, and it reveals that mackerel achieved the highest average precision (AP) value of 0.927, whereas sardine achieved the lowest average precision value of 0.740. Considering all classes, the mAP value was 0.835.

The models for this plant were trained using the Google Colab platform, with a total time of 1.358 h to complete 150 epochs in training, using an Nvidia A100 GPU. The inference time obtained while processing a whole image was 18.5 ms, which corresponds to a frame rate of 54 FPS.

### 3.2. Results for Plant 2

For Plant 2, the model was evaluated for the detection of jack mackerel, mackerel, jumbo squid, and snoek using a total of 1771 validation samples. The validation results are summarized in the confusion matrix in Figure 8. Here, it can be observed that an accuracy of 0.98 was achieved for jack mackerel, 0.84 for mackerel, 1.0 for jumbo squid, and 0.4 for snoek, resulting in a value of 0.805 for MP. On the one hand, the FP background precision value was only 0.01 for jack mackerel and 0 for the rest of the fish classes. On the other hand, the maximum FN background precision value was 0.52 for jack mackerel and the minimum FN background precision value was 0 for jumbo squid.

Figure 9a presents the evolution of the loss function and mAP during the training process of the model in Plant 2. The red dot on the plot indicates epoch 97, which corresponds to the weights used with the model to achieve the best results. At this epoch, the model achieved a maximum mAP value of 0.88359, indicating its optimal performance in terms of detection precision.

The precision–recall curve of the model in this plant is shown in Figure 9b. On the one hand, the maximum AP value of 0.995 was achieved for jumbo squid, indicating high precision in detecting this class. On the other hand, snoek achieved the lowest AP value of 0.795, suggesting room for improvement in its detection performance, as it was observed in the confusion matrix during validation, where the model may have struggled to accurately detect and classify this particular species. Considering all classes, the mean average precision (mAP) value was 0.888, representing an overall measure of the model’s performance.

The models for this plant were also trained using the Google Colab platform, with a total time of 1.6 h in completing 150 epochs in training, using an Nvidia A100 GPU. The inference time obtained while processing a whole image was 18.5 ms, which corresponds to a frame rate of 54 FPS.

### 3.3. Discussion

In terms of model performance, the results from Plant 1 indicate that the accuracy for larger species was generally higher. This can be attributed to the fact that the cameras are capable of capturing greater detail and distinguishing the characteristics of larger fish species more effectively compared to smaller ones. In Figure 10a, we can see the correct detection of a mackerel (indicated by the violet bounding box and mask) and two jack mackerels (indicated by the lime green bounding box and mask), demonstrating that these large species were detected accurately without any issues. The most challenging species for the system were the smaller species, such as sardine and anchovy. In Figure 10b, the detection of a sardine is shown. With smaller species like sardine and anchovy, it was common to encounter images where a significant number of these species were present, making it difficult to correctly identify and classify all of them. To address these difficulties, further optimization and fine-tuning of the model may be necessary, taking into consideration the specific characteristics and behaviors of smaller species. Additionally, exploring advanced techniques, such as data augmentation, ensemble methods, or leveraging additional training data focused on smaller species, can potentially improve the system’s performance in accurately detecting and classifying these challenging scenarios.

In Figure 10, the functionality of the implemented length and weight estimation module is also showcased, with the estimated values displayed beneath each bounding box. This module provides real-time approximations of the length and weight of the detected species. However, there are certain challenges that still need to be addressed. For instance, the orientation of the fish and whether the model detects the entire fish or only a portion of it can affect the accuracy of the estimations. To overcome these challenges, there are plans to incorporate an additional model in the future. This model will randomly verify the detections to ensure that they correspond to a complete fish. This approach will help improve the accuracy of the length and weight estimations by focusing on reliable and complete fish detections. The use of a key-point recognition model has also been proposed for the accurate identification of the head and tail end of the species to determine its standard length instead of its total length.

The results obtained from Plant 2 highlight the challenges in snoek detection, as visually distinguishing snoek from mackerel and jack mackerel was not straightforward from our imagery. Nevertheless, jumbo squid was easily recognizable by the system due to its large size and distinctive color, making it stand out compared to other species.

In Figure 11a, an example of identification can be observed, with a significant presence of jack mackerel (indicated by the yellow bounding boxes) on the conveyor belt. On the right side of the image, two mackerels are identified (indicated by the red bounding box), characterized by a pattern of irregular black striations on their dorsal area. On the left side of the image, a snoek is detected (indicated by the green bounding box). It should be noted that the snoek species is larger compared to the other species, but it may not appear completely within the image capture. Figure 11b displays the detection of a jumbo squid (indicated by the orange bounding box) with some jack mackerel around it. Due to its size, which is similar to snoek, it is unlikely to be captured entirely in a single image with the current zoom configuration. Additionally, jumbo squid are known to hide under jack mackerel, making their visible size variable.

In terms of practical considerations, when using YOLOv7 for real-time classification of fish species with varying sizes, there is a delicate balance between the frame rate and image resolution. In our case, we adjusted the frame rate to synchronize with the conveyor belt’s speed, ensuring that the same fish did not appear in two consecutive images. This simple optimization allows for the efficient utilization of computing resources to process multiple images simultaneously.

Furthermore, image resolution plays a crucial role in distinguishing smaller or similarly-sized species such as sardine or anchovy. We decided to maintain YOLOv7’s default input image resolution of 640×640 pixels. This choice resulted in an inference frame rate 6.75 times faster than the camera frame rate at Plant 1 and 54 times faster than the camera frame rate at Plant 2. Opting for a higher resolution could potentially enhance feature extraction capabilities but at the cost of reduced frame rates, thereby limiting the ability of the system to process multiple images from the same plant concurrently.

As shown in the results, the system allows for the identification of species with high accuracy. Nevertheless, there are certain inherent limitations in the methodology. The system can only detect species on the top layer, is sensitive to overlapping objects, and fails to classify fish in a deteriorated state. Additionally, some environmental sources of error such as intense sunlight can cause pixel saturation, as well as humidity conditions, which may result in fogging, requiring the system to be cleaned by plant operators.

## 4. Conclusions

In this work, we have presented a computer vision system for fish detection, classification, and weight estimation, fine-tuned for two different landing plants of Chilean pelagic fish. The obtained results demonstrate the system’s ability to detect and distinguish different species, as well as the challenges associated with certain species that exhibit visual similarities and the limitations in capturing them completely in a single image.

The implementation of the system represents a significant improvement compared to the current inspection procedures, providing an automated method for the identification and estimation of the size and weight of fish. The system is able to provide more accurate and reliable fish landing traceability to fisheries management agencies, which could contribute to the sustainability of marine resources and the long-term viability of the fishing industry. Further work includes extending the current database to incorporate new marine species, as well as improving the performance of the weight and length estimation module.

## Figures and Tables

**Figure 1 sensors-23-08163-f001:**
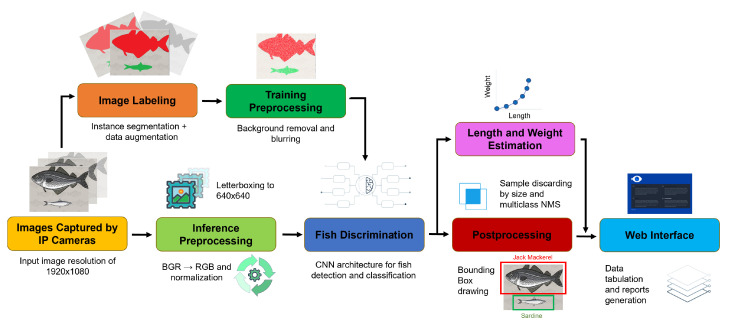
Architecture of the fish recognition system.

**Figure 2 sensors-23-08163-f002:**
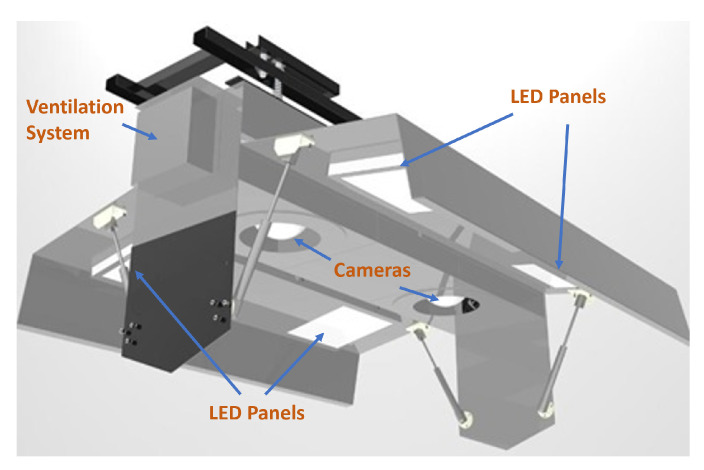
Schematic of gate installed in Plant 1.

**Figure 3 sensors-23-08163-f003:**
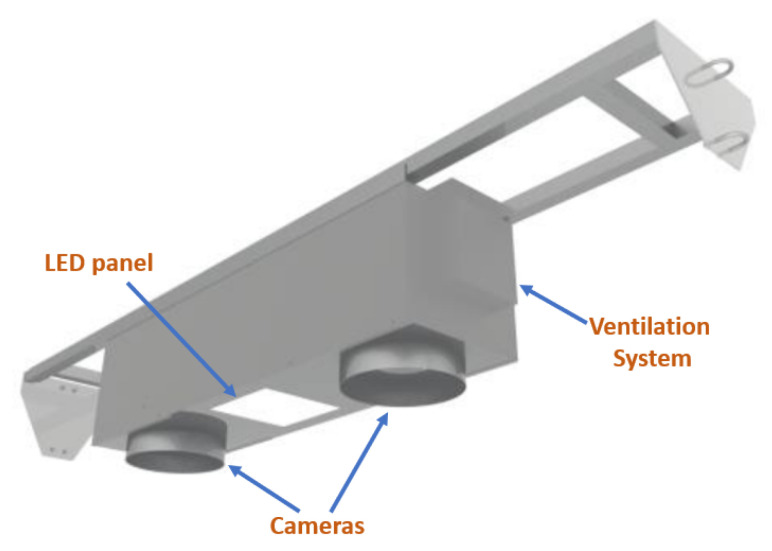
Schematic of gate installed in Plant 2.

**Figure 4 sensors-23-08163-f004:**
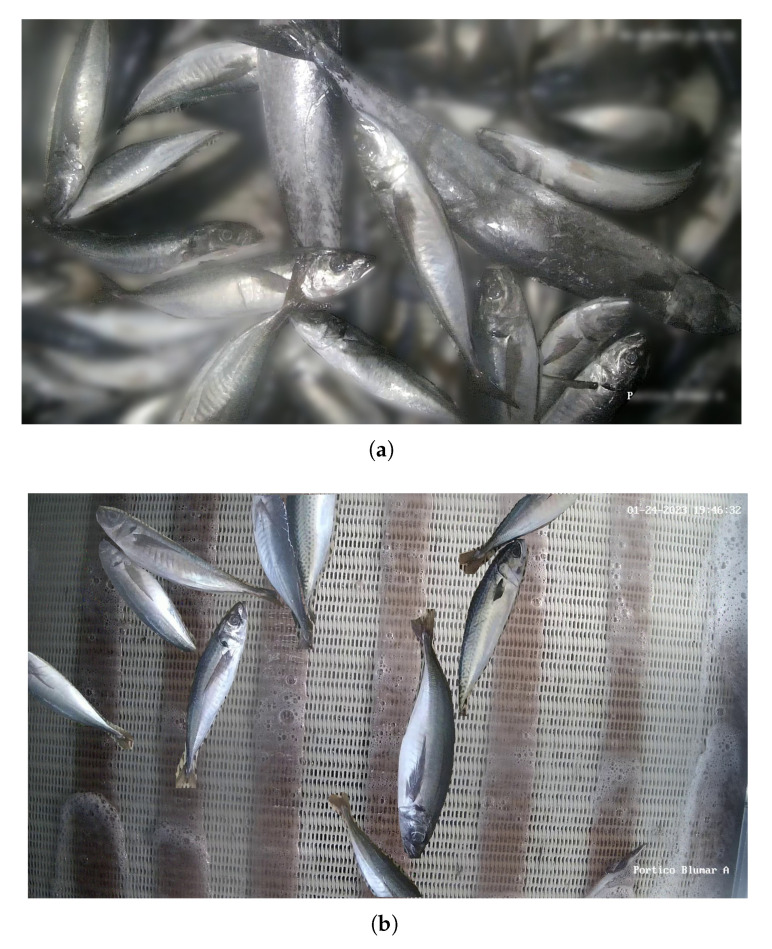
Modified images for database construction. (**a**) Modified image with blurring effect. (**b**) Modified image with random background.

**Figure 5 sensors-23-08163-f005:**
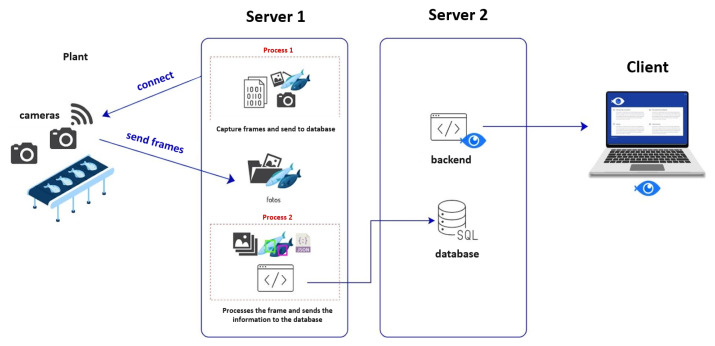
Developed web interface diagram.

**Figure 6 sensors-23-08163-f006:**
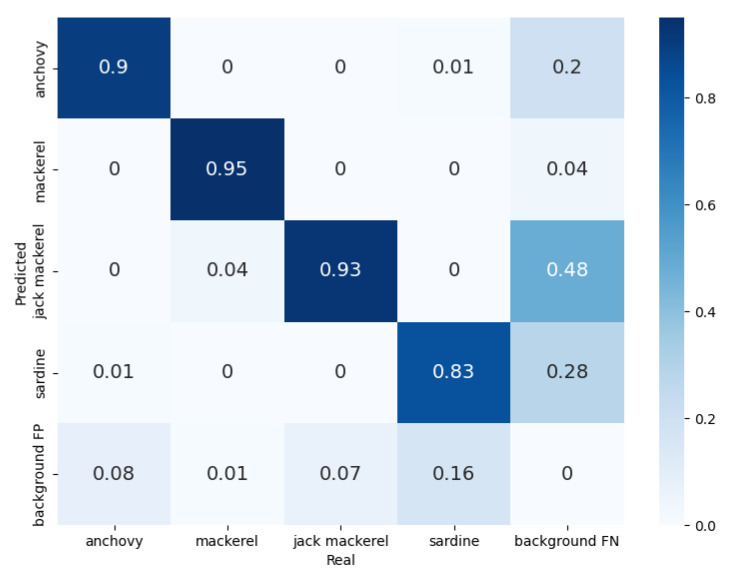
Confusion matrix obtained during the validation of the model for Plant 1.

**Figure 7 sensors-23-08163-f007:**
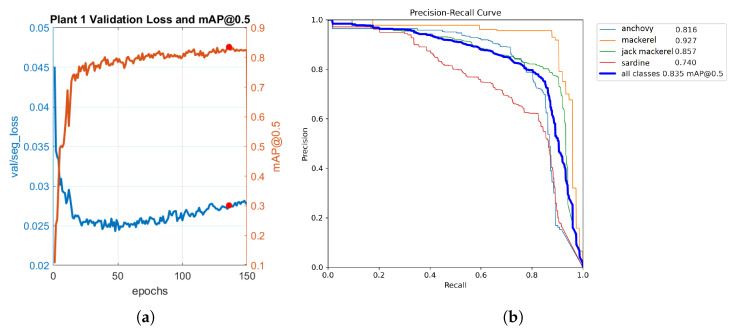
Results for Plant 1. (**a**) Evolution of loss function and mAP of the YOLOv7 model. (**b**) Precision–recall curve for YOLOv7 mask outputs.

**Figure 8 sensors-23-08163-f008:**
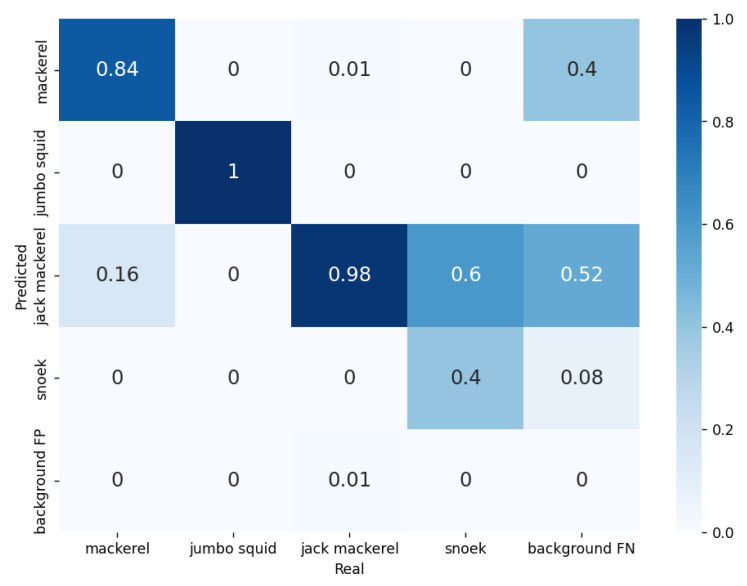
Confusion matrix obtained during the validation of the model for Plant 2.

**Figure 9 sensors-23-08163-f009:**
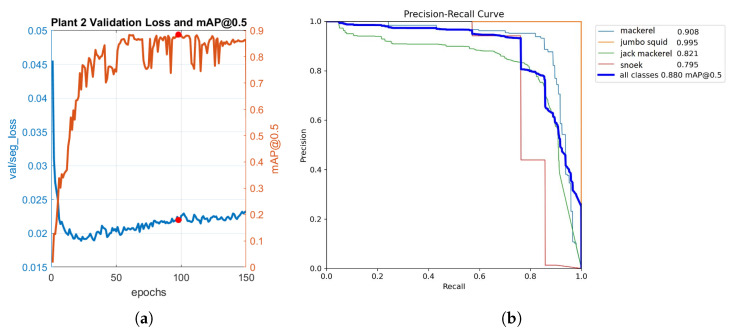
Results for Plant 2. (**a**) Evolution of loss function and mAP of the YOLOv7 model. (**b**) Precision–recall curve for YOLOv7 mask outputs.

**Figure 10 sensors-23-08163-f010:**
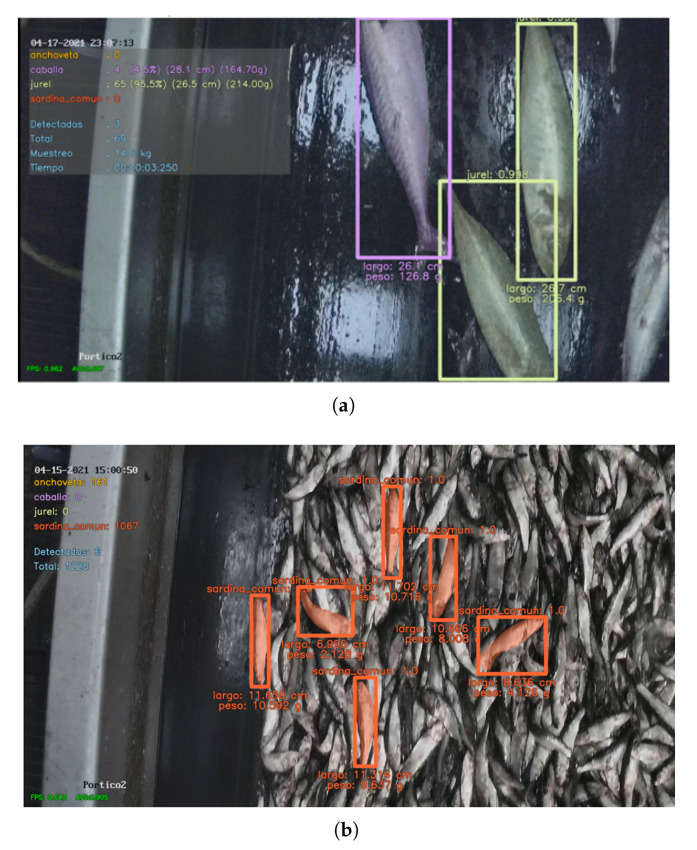
Species detections in Plant 1. (**a**) Large species. (**b**) Small species.

**Figure 11 sensors-23-08163-f011:**
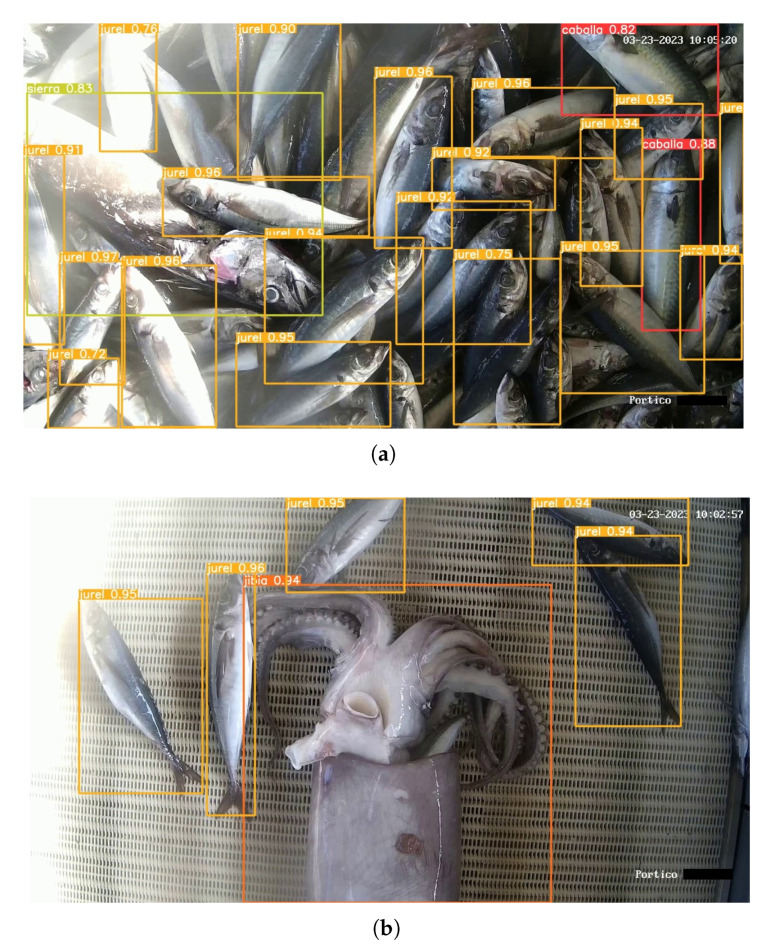
Species detections in Plant 2. (**a**) Detection of several species simultaneously. (**b**) Detection of jumbo squid and jack mackerel.

**Table 1 sensors-23-08163-t001:** Database of labeled samples for each fishing plant.

Class	Samples (Plant 1)	Samples (Plant 2)
Anchovy	549	-
Jack Mackerel	1339	7377
Mackerel	139	764
Sardine	480	-
Jumbo Squid	-	22
Snoek	-	177

**Table 2 sensors-23-08163-t002:** Weight and length estimation parameters.

Classes	a	b
Anchovy	0.003653	3.23344
Mackerel	0.0090147	3.1067
Jack Mackerel	0.007921	3.1085
Sardine	0.0064065	3.1299

## Data Availability

Not applicable.

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
