# Peer review of "Digital Classification of Chilean Pelagic Species in Fishing Landing Lines"

_sensors, 2023, doi:10.3390/s23198163_

Round 1

Reviewer 1 Report

1) Typos exist. " similar conditions than the first one" (Line 115) should be "similar to". "Data was captured" should be plural form "were".

2) Amending specific information is critical to clarify some descriptions, for example, the positions of cameras and LEDs are missing in Figure 3. In addition, there is a lack of data acquisition year (Line 126-131). In the pre-processing, replacement of unlabeled areas method should be explained in detail (Line 147-153).

3) Difference between YOLO v4 and YOLO v8 has been discussed, however, more comparisons are needed in terms of other YOLO models, especially the latest YOLO v8.

4) Precision is defined by decimal while MP turns to be a percentage value. It is better to remain the same digit format (Line 313-320).

5) In the discussion part, the lower accuracy of several species have been analysed. As YOLO algorithms is based on the end-to-end classification structure, it is necessary to provide the intermediate outcomes of different layers for amending the comparison.

Language quality is fine except for several typos.

Reviewer 2 Report

Title and Abstract:

The title accurately represents the content of the article. The abstract provides a clear overview of the challenges in the Chilean fishing industry's inspection process and introduces the proposed automated fish landings control system. However, it would be beneficial to include a brief mention of the main findings or results achieved through this system.

Introduction:

It might be helpful to provide some quantitative data on the scale of the challenge, such as the number of artisan vessels or the volume of landings.

Related Work:

The integration of deep learning techniques like YOLOv4, YOLOv7, and MASK R-CNN is well-highlighted. However, it might be useful to provide a brief description of each of these techniques for readers who might not be familiar with them.

Practical Insights:

Whenever applicable, provide insights into practical considerations that influenced decisions. For example, discuss the trade-offs considered when selecting the image capture rate or the frame resolution, and explain how these decisions impacted the system's real-time performance.

Address Limitations:

Include a brief section addressing any limitations of the methodology. Discuss potential sources of error, such as variations in lighting conditions during data capture or challenges in accurately labeling overlapping fish instances.

It's advisable to consider having your work proofread for improved accuracy and quality.
